

# Mobile learning in dentistry: usage habits, attitudes and perceptions of undergraduate students

Aslı Suner[1], Yusuf Yilmaz[2] and Beyser Pişkin[3]

[1] Department of Biostatistics and Medical Informatics/Faculty of Medicine, Ege University, İzmir, Turkey
[2] Department of Medical Education/Faculty of Medicine, Ege University, İzmir, Turkey
[3] Division of Endodontology/School of Dentistry, Ege University, İzmir, Turkey

## ABSTRACT

**Introduction.** The aim of this study was to evaluate usage habits, attitudes and perceptions towards mobile learning (m-learning), as well as to identify variables related to those attitudes amongst undergraduate dental students.

**Materials and Methods**. The study consists of 81 dental undergraduate students who who volunteered to participate. The data collection tool consists of an m-learning attitude scale, a questionnaire, and open-ended questions. To compare the total scores and factors of m-learning attitude scale for demographic information and mobile technology usage habits of the students; the Mann–Whitney $U$ test was used for two independent groups such as gender, presence of electronic devices, and places of Internet usage. The Kruskal–Wallis test was also used to compare the total scores and factors of m-learning attitude scale for more than two independent groups including internet usage purposes and opinions. Spearman's correlation coefficient was performed, and linear regression analysis was used to predict the change in total score according to the purposes of Internet usage.

**Results**. The majority of students thought that the use of mobile devices in dentistry courses was useful and their attitudes towards m-learning were high. The students generally use the Internet for online shopping, connecting to social networks, and communication. They tend to use mobile technologies for personal use, followed by educational purposes. There were significant differences found in the m-learning attitudes for gender, having a portable power supply and use of mobile devices in dentistry courses. Communication was found significant in predicting the change in total score for the m-learning attitude scale according to the purpose of Internet usage.

**Conclusion**. Dental students have generally positive attitudes towards m-learning. Students raise awareness towards the promise of m-learning in order to apply their individual technology use and learning behaviours. Designing learning materials and applications for mobile devices may increase students' performances.

Corresponding author
Yusuf Yilmaz,
yusuf.yilmaz@ege.edu.tr,
yusufyyilmaz@gmail.com

# INTRODUCTION

Since *Mattheos et al. (2008)* first identified information technology (IT) related activities for use in dental education, technology has increasingly developed, both in terms of
hardware and software applications. The use of mobile technologies in daily life has become widespread, and equally, educational technologies have taken advantage of this opportunity. Learning environments can be supported with the help of mobile technologies such as smartphones, tablet computers and laptop computers (*Elçiçek & Bahçeci, 2017*). Specifically, learning on mobile devices, which is termed as mobile learning (m-learning) is defined as the use of mobile technologies in educational activities (*Sharples, 2013*; *Stone, 2004*; *Winters, 2006*). Learning through mobile devices can be achieved ubiquitously, through the provision of instant access to information (*Sharples, 2000*). Results of m-learning research emphasize that educational activities lead to more meaningful learning where learning contents are designed appropriately according to learners' interests and needs (*Elçiçek & Bahçeci, 2017*).

There are multiple advantages that m-learning brings to education. First of all, since educational programmes loaded with time-intensive and knowledge-intensive courses are becoming more and more difficult for students to prepare for each lecture, the efficiency of such programmes become critically affected (*Harvey, Rothman & Frecker, 2003*). Second, whilst it is possible for students to withstand intensive course content delivery over a short period of time, prolonged instruction at this intensity may make meaningful learning more difficult (*Van Merriënboer & Sweller, 2010*). Third, in education programmes where Internet technologies are used, solutions are introduced that employ various approaches. The widespread usage of mobile technologies today makes m-learning even more prominent. However, each student's access to mobile technologies, and their level and purpose of usage may of course differ. With a general 'Bring Your Own Device' (BYOD) policy in higher education, m-learning activities are reliant upon usage of technology equipment owned by the students (*Hwang, Lai & Wang, 2015*; *Song, 2014*). In particular, knowing what kind of tools and usage habits students have, and then designing educational activities accordingly can go some way to the provision of equal opportunities in education (*Şahin & Başak, 2017*). Last but not least, the perspectives of students towards educational activities with such technologies can significantly affect their attitudes towards learning (*Yağcı, 2017*).

Nowadays, most dental students are from Generation Z. Although this generation was born into technology and with technological devices very much a part of their daily life; it does not necessarily mean that they can or want to learn with and on mobile devices (*Khatoon, Hill & Walmsley, 2014*; *The Center for Generational Kinetics, 2018*). Therefore, it is important to understand dental students' attitudes and opinions towards m-learning.

Use of mobile technologies has dramatically increased in the last decade (*Fu & Hwang, 2018*). M-learning has attracted attention in many fields such as engineering (*Humanante-Ramos, García-Peñalvo & Conde-González, 2017*), teacher education (*Dashtestani, 2016*), and business management (*Al-Emran, Elsherif & Shaalan, 2016*). Mobile technologies are also seen as increasingly preferred for use by researchers in health sciences including medicine, pharmacology, and nursing in order to educate students as well as communication for the purposes of consultation and for accessing scientific research in support of patient care (*Al-Emran, Elsherif & Shaalan, 2016*; *Briz-Ponce et al., 2016*; *Cain, Bird & Jones, 2008*; *Davies et al., 2012*; *Rodis et al., 2016*). While attitudes of dental students

are generally positive towards e-learning (*Brumini et al., 2014*), their attitudes towards m-learning were not investigated to our knowledge. However higher education students and medical students were included in previous studies to explore attitudes or perceptions towards m-learning based on age, country, gender, smart-phone usage and class (*Al-Emran, Elsherif & Shaalan, 2016*; *Patil et al., 2016*). Although dentistry-related mobile applications are becoming very popular (*Khatoon, Hill & Walmsley, 2014*), less is known about the usage habits, attitudes, opinions of dental students, or their perceptions and approaches to change in terms of m-learning. Thus, the researchers of the current study believe that determining the mobile technology usage habits of undergraduate dental students, and examining their attitudes and perceptions towards m-learning may contribute to the development of m-learning enabled educational programmes.

Therefore, the aim of the current study was to determine the mobile technology usage habits of 81 undergraduate dental students and their attitudes and opinions towards m-learning in terms of various variables. Specifically, the study objectives were as follows: (1) to describe usage of mobile technology in dental students, (2) to explore students attitudes towards m-learning, and (3) to explore the role of the students' demographic information, and the purpose and habits of using mobile technologies as examined by previous research studies (*Deshpande, Kalaskar & Chahande, 2016*; *Khatoon, Hill & Walmsley, 2014*; *Rung, Warnke & Mattheos, 2014*).

# MATERIALS AND METHODS

To address the aim of the study, a cross-sectional survey design, which is used to collect information at a specific period of time (*Fraenkel, Wallen & Hyun, 2014*), was utilized. Ethics committee approval was received from Ege University Scientific Research and Publication Ethics Health Sciences Board with 145-2018 reference number at 15 May 2018 prior to the study's commencement.

## Sample

The sample consisted of 81 undergraduate students, from 1st to 5th years, who volunteered to participate to the study during the Spring Semester of the 2017–2018 Academic Year at Ege University's Faculty of Dentistry, Turkey. Since all of the population was intended to be reached, no sampling method was determined or applied.

## Instruments

The data collection tool consisted of three parts, an m-learning attitude scale, a questionnaire, and open-ended questions (File S1).

### M-Learning attitude scale

This scale, which was developed by *Çelik (2013)* and designed as a five-point, Likert-type scale instrument, consists of 21 items, of which five are reverse-scored items. The total score for the scale ranges from 21 to 105. The higher the total score, the greater the positive attitude towards m-learning. In order to determine the validity of the scale, the Cronbach's alpha coefficient was calculated as 0.881 (*Çelik, 2013*). There are four factors in the scale: 'Advantages of M-Learning (F1)', 'Limitations in M-Learning (F2)', 'Usability

in M-Learning (F3)', and 'Freedom in M-Learning (F4)'. There are five different response options levels for each question: '5-Strongly Agree', '4-Agree', '3-Undecided', '2-Disagree', and '1-Strongly Disagree'. In the current study, the Cronbach Alpha coefficient of the scale was calculated as 0.886, and the scale was found to be highly reliable.

### Student information form

This questionnaire was prepared by the researchers based on previous research studies in the literature, and contains a total of 11 questions based on the respondent's demographic information (gender, year of birth, etc.) and their mobile technology usage habits (presence of electronic devices, places of Internet usage, Internet usage purposes, and opinions, etc.) in general (*Deshpande, Kalaskar & Chahande, 2016*; *Khatoon, Hill & Walmsley, 2014*; *Rung, Warnke & Mattheos, 2014*).

### Opinions about mobile learning

This part of the questionnaire consists of five open-ended questions which the researchers of the study generated, and can be seen as below. Whilst not obligatory to answer, the aim of these questions was to gather general views of the students about m-learning.

1. Are there any mobile phone applications that you use for training purposes? Please indicate if applicable.
2. What are the positive aspects of course learning with mobile learning?
3. What are the negative aspects of course learning with mobile learning?
4. What are your expectations for the use of mobile tools in the processing of courses?
5. What kind of content do you want to see on mobile devices to increase your success in your courses?

## Data collection and analysis

The data collection instruments were transferred to LimeSurvey which was used as an online data collection system (*LimeSurvey, 2018*) for this study and placed on the faculty website for the purposes of data collection (https://okm.med.ege.edu.tr/anket/index.php?r=survey/index&sid=468989). This weblink was announced to the target student population via the Ege University Faculty of Dentistry Learning Resources Centre, and students were asked to participate in the survey voluntarily. In addition, since one of the researchers was a lecturer in the Faculty of Dentistry, and the other researcher was the Dean of the Faculty, the survey was also verbally announced to the students and that voluntary participation was anticipated. Moreover, announcements were made through student communication groups (e.g., WhatsApp), and via student noticeboards located in the faculty.

With the consent form on the opening page of the online questionnaire, prospective participant students were informed in more detail about the study and their consent to participate was obtained. The data collected was stored within a secure database that was only accessible to the researchers. The data collection process lasted for a period of two weeks. In order to increase participation in the study, announcements were repeated several times during the data collection period.

While continuous variables were presented as median (min–max), categorical variables were described with frequencies and percentages. Cronbach's alpha for the m-learning

attitude scale was calculated. Shapiro–Wilk normality test was used to examine whether the numerical data were distributed normally. Since the data were not found to be normally distributed, to compare the total scores and factors of m-learning attitude scale for demographic information and mobile technology usage habits of the students, Mann–Whitney $U$ test was used for two independent groups such as gender, presence of electronic devices, and places of Internet usage. Kruskal-Wallis test followed by Mann–Whitney $U$ test was applied in order to see the *post-hoc* results of pairwise *comparisons* was also used to compare the total scores and factors of m-learning attitude scale for more than two independent groups including internet usage purposes and opinions. Correlation analyses were performed using Spearman's correlation coefficient. A stepwise linear regression analysis was used in order to construct a model that predicted the change in total score of the scale according to purpose of Internet usage. A final model was executed on all the variables for the purpose of Internet usage, and they were included together in the model. A value of $P < .05$ was considered statistically significant. Statistical analyses were performed using IBM SPSS version 25.0 statistical software for personal computers.

## RESULTS

The results of the study were presented in four sections. First, the overall m-learning attitudes of the participant students were summarised. Second, demographic information, mobile technology usage habits gathered from the student information form, and the total score from the m-learning attitude scale were described and compared. Next, significant variables were examined based on each factor of the m-learning attitude scale. Third, Internet usage purposes were modelled in order to predict how the m-learning attitudes of students changed. Finally, opinions of students about m-learning were reported.

### Attitudes of dental students toward M-Learning

The factors of the scale, 'Advantages of M-Learning', 'Limitations in M-Learning', 'Usability in M-Learning', and 'Freedom in M-Learning' were examined. The results demonstrated that students generally reported positive opinions in respect of m-learning. The median, minimum and maximum values of the m-learning attitude scale factors and total score are presented in Table 1. The mean scores of the factors were found to be high, and the mean total score of the scale was $79.52 \pm 10.62$ which revealed that the dentistry students have a high attitude towards m-learning.

### Demographics and M-Learning attitude

Of the 81 student participants, 55.6% ($n = 45$) were female. The total score of the male participants was found to be statistically significantly higher than that of the females ($P = 0.025$). The average age of the students was 21.23 years $\pm 1.56$.

The results showed that 97.5% ($n = 79$) of the students had a smartphone, 23.5% ($n = 19$) had a tablet computer and 85.2% ($n = 69$) had a laptop computer. There was no statistically significant difference found for the overall scores ($P = 0.110$, $P = 0.584$, $P = 0.283$, respectively) among smartphone, tablet, and laptop owners. In terms of their Internet connectedness, 98.8% of the students ($n = 80$) had an average of $6.38 \pm 3.69$

**Table 1  M-Learning attitude scale factors and total score.**

| Factors | Median | Min–Max | Original scale Min–Max |
|---|---|---|---|
| F1: Advantages of M-Learning | 26.00 | 14.00–35.00 | 7.00–35.00 |
| F2: Limitations in M-Learning | 17.00 | 5.00–25.00 | 5.00–25.00 |
| F3: Usability in M-Learning | 20.00 | 13.00–25.00 | 5.00–25.00 |
| F4: Freedom in M-Learning | 16.00 | 11.00–20.00 | 4.00–20.00 |
| Total score | 79.00 | 58.00–101.00 | 21.00–105.00 |

GB Internet package. While 38.3% of the 81 students ($n = 31$) carried a charger with them, 25.9% ($n = 21$) had a portable power supply unit. While there was no statistically significant difference found between the total scores of the scale and those without chargers ($P = 0.214$), those with a portable power supply unit were found to have statistically higher total scores than those who did not ($P = 0.019$).

In terms of their smartphone usage, 91.4% ($n = 74$) of the students reported that they checked their mobile phones when they woke up; while 97.5% ($n = 79$) checked their mobile phones before going to sleep. There was no statistically significant difference found between the total scores of the students who checked their phone and those who did not when either they woke up or before going to sleep ($P = 0.060$, $P = 0.837$, respectively).

For the students Internet access, 85.2% of the students ($n = 69$) accessed the Internet at home, 44.4% ($n = 36$) accessed the Internet from their student dormitories, 79.0% ($n = 64$) used the Internet at school, 95.1% ($n = 77$) reached the Internet using their mobile phone, 61.7% ($n = 50$) accessed the Internet in cafe-restaurants, and 46.9% ($n = 38$) used free public Wi-Fi Internet sources for $4.49 \pm 2.29$ h per day. There was no statistically significant difference found between the overall scores of the scale in terms of Internet access locations ($P > 0.05$).

When the students were asked to list the purposes of their Internet usage, the distribution of options from the greatest to the least were online shopping ($n = 17$, 22.1%), connecting to social networks ($n = 17$, 22.1%), communicating ($n = 13$, 16.9%), watching videos ($n = 7$, 9.1%), listening to music ($n = 7$, 9.1%), reading scientific articles ($n = 4$, 5.2%), accessing course materials ($n = 4$, 5.2%), reading the news ($n = 4$, 5.2%), checking e-mails ($n = 2$, 2.6%), and playing games ($n = 2$, 2.6%). No statistically significant difference was found between the total scores of the scale in terms of purposes of Internet usage ($P = 0.233$).

With regard to their courses, 51.4% of the students ($n = 39$) responded that use of mobile devices in their dentistry courses was deemed to be useful, while 35.5% ($n = 27$) responded that it was very useful. None of the students stated that it was not useful at all. A statistically significant difference was found between the opinions of the students' usage of mobile devices in the courses given in dentistry ($P < 0.001$). Pairwise comparisons were made in order to see the post-hoc results, and statistically significant differences were found between the responses of 'undecided' and 'agreed', between 'undecided' and 'strongly agreed', and between 'agreed' and 'strongly agreed' ($P = 0.004$; $P < 0.001$; $P = 0.001$, respectively).

**Table 2 Demographic information and mobile technology usage habits of the students in terms of the m-learning attitude scale total scores.**

| Variable | | n (%) | Total score median (Min–Max) | P-value |
|---|---|---|---|---|
| **Gender** | **Female** | 45 (55.60) | 78.50 (61.00–92.00) | 0.025[a] |
| | **Male** | 36 (44.40) | 82.50 (62.00–101.00) | |
| **Age (years)** | **Median (Min–Max)** 21 (19–25) | – | – | – |
| **Smartphone** | + | 79 (97.50) | 79.50 (61.00–101.00) | 0.110[a] |
| | – | 2 (2.50) | 91.00 (85.00–97.00) | |
| **Tablet computer** | + | 19 (23.50) | 79.00 (62.00–101.00) | 0.584[a] |
| | – | 62 (76.50) | 80.00 (61.00–100.00) | |
| **Laptop computer** | + | 69 (85.20) | 80.50 (61.00–101.00) | 0.283[a] |
| | – | 12 (14.80) | 76.00 (62.00–94.00) | |
| **Mobile internet package** | + | 80 (98.80) | 80.00 (61.00–101.00) | – |
| | – | 1 (1.20) | – | |
| **Mobile internet package size (GB)** | **Median (Min–Max)** 6 (1–25) | – | – | – |
| **Carry a charger unit** | + | 31 (38.30) | 81.00 (62.00–100.00) | 0.214[a] |
| | – | 50 (61.70) | 79.00 (61.00–101.00) | |
| **Portable power supply** | + | 21 (25.90) | 84.00 (65.00–100.00) | 0.019[a] |
| | – | 60 (74.10) | 79.00 (61.00–101.00) | |
| **Checking mobile phone when waking up** | + | 74 (91.40) | 81.00 (61.00–101.00) | 0.060[a] |
| | – | 7 (8.60) | 70.00 (62.00–86.00) | |
| **Checking mobile phone before going to sleep** | + | 79 (97.50) | 80.00 (61.00–101.00) | 0.837[a] |
| | – | 2 (2.50) | 80.50 (79.00–82.00) | |
| **Using internet at home** | + | 69 (85.20) | 80.50 (61.00–101.00) | 0.434[a] |
| | – | 12 (14.80) | 76.50 (62.00–100.00) | |
| **Using internet in student dormitory** | + | 36 (44.40) | 77.00 (61.00–100.00) | 0.083[a] |
| | – | 45 (55.60) | 80.00 (62.00–101.00) | |
| **Using internet in school** | + | 64 (79.00) | 80.00 (61.00–101.00) | 0.971[a] |
| | – | 17 (21.00) | 79.00 (64.00–97.00) | |
| **Using internet with a mobile phone** | + | 77 (95.10) | 79.50 (61.00–101.00) | 0.427[a] |
| | – | 4 (4.90) | 82.50 (77.00–92.00) | |
| **Using internet in cafes-restaurants** | + | 50 (61.70) | 81.00 (61.00–101.00) | 0.162[a] |
| | – | 31 (38.30) | 79.00 (62.00–100.00) | |
| **Using free public Wi-Fi internet** | + | 38 (46.90) | 81.00 (61.00–101.00) | 0.578[a] |
| | – | 43 (53.10) | 79.00 (62.00–100.00) | |

**Table 2** (*continued*)

| Variable | | *n* (%) | Total score median (Min–Max) | *P*-value |
|---|---|---|---|---|
| **Average daily internet usage (hours)** | **Median (Min–Max)** 4.00 (1.00–12.00) | – | – | – |
| **Internet usage purposes** | **Online shopping** | 17 (22.10) | 82.00 (61.00–101.00) | |
| | **Connecting to social networks** | 17 (22.10) | 80.00 (62.00–97.00) | |
| | **Watching videos** | 7 (9.10) | 81.00 (62.00–99.00) | |
| | **Reading scientific articles** | 4 (5.20) | 88.00 (82.00–97.00) | |
| | **Accessing course materials** | 4 (5.20) | 85.50 (72.00–94.00) | 0.233[b] |
| | **Checking e-Mails** | 2 (2.60) | 88.00 (76.00–100.00) | |
| | **Reading the news** | 4 (5.20) | 81.00 (80.00–92.00) | |
| | **Communicating** | 13 (16.90) | 75.00 (62.00–84.00) | |
| | **Listening to music** | 7 (9.10) | 79.00 (70.00–92.00) | |
| | **Playing games** | 2 (2.60) | 71.00 (63.00–79.00) | |
| **Use of mobile devices in dentistry courses is beneficial** | **Strongly Agreed** | 27 (35.50) | 85.50 (64.00–101.00) | |
| | **Agreed** | 39 (51.40) | 79.00 (61.00–94.00) | |
| | **Undecided** | 8 (10.50) | 67.50 (62.00–79.00) | <0.001[b] |
| | **Disagreed** | 2 (2.60) | 73.50 (63.00–84.00) | |
| | **Strongly disagreed** | – | – | |

Notes.
[+]Present.
[–]Absent.
[a]Mann–Whitney *U* Test.
[b]Kruskal Wallis Test.

As can be seen in Table 2, participants' demographic information and mobile technology usage habits were examined in terms of the m-learning attitude scale total scores. In Table 3, the score of the second factor for males was found to be statistically significant and higher than that of the females ($P = 0.018$).

Students who carried portable power supply units scored statistically higher in the third and fourth factors of the scale compared to those who did not carry power supply units ($P = 0.015$, and $P = 0.024$, respectively). There were statistically significant differences between the student opinions for the use of mobile devices in dentistry courses in terms of the four factor scores ($P = 0.020$; $P = 0.009$; $P = 0.003$; and $P < 0.001$; respectively). According to the results of pairwise comparisons, the score of all four factors for students who 'strongly agreed' were found to be statistically significant and higher than students who 'agreed' or were 'undecided' ($P < 0.05$).

In Table 4, a statistically significant correlation was found between average daily Internet usage time and the second factor of the m-learning attitude scale, which represents the perceived limitations of m-learning ($r = 0.243$; $P = 0.032$). Similarly, a significant correlation was found to exist between the age of dentistry students and the third factor of the m-learning attitude scale named 'Usability in m-learning', and the total score of the scale ($r = 0.291$; $P = 0.010$, and $r = 0.229$; $P = 0.044$, respectively).

## Effects of internet usage purposes on M-Learning

A linear regression analysis was used to predict the change in total score according to the purposes of Internet usage. Therefore, neither sociodemographic information nor

**Table 3 Scores of M-Learning attitude scale factors for statistically significant variables.**

| Variable | | F1 Median (Min–Max) | P-value | F2 Median (Min–Max) | P-value | F3 Median (Min–Max) | P-value | F4 Median (Min–Max) | P-value |
|---|---|---|---|---|---|---|---|---|---|
| **Gender** | **Female** | 26.00 (16.00–35.00) | 0.078 | 16.50 (5.00–22.00) | 0.018[*] | 20.00 (13.00–25.00) | 0.414 | 16.00 (11.00–20.00) | 0.094 |
| | **Male** | 27.50 (20.00–35.00) | | 18.50 (10.00–25.00) | | 21.00 (15.00–25.00) | | 16.50 (12.00–20.00) | |
| **Portable power supply** | **+** | 27.00 (23.00–35.00) | 0.143 | 18.00 (5.00–24.00) | 0.179 | 21.00 (15.00–25.00) | 0.015[*] | 17.00 (14.00–20.00) | 0.024[*] |
| | **−** | 26.00 (16.00–35.00) | | 17.00 (10.00–25.00) | | 20.00 (13.00–25.00) | | 16.00 (11.00–20.00) | |
| **Use of mobile device in dentistry courses is beneficial** | **Strongly agreed** | 28.50 (22.00–35.00) | | 19.00 (11.00–25.00) | | 22.00 (15.00–25.00) | | 18.00 (14.00–20.00) | |
| | **Agreed** | 26.00 (20.00–35.00) | 0.020[*] | 17.00 (11.00–22.00) | 0.009[*] | 20.00 (13.00–25.00) | 0.003[*] | 16.00 (12.00–20.00) | <0.001[*] |
| | **Undecided** | 23.00 (16.00–31.00) | | 14.00 (10.00–20.00) | | 18.00 (14.00–21.00) | | 14.00 (11.00–16.00) | |
| | **Disagreed** | 25.00 (22.00–28.00) | | 14.50 (11.00–18.00) | | 18.50 (16.00–21.00) | | 15.50 (14.00–17.00) | |
| | **Strongly disagreed** | – | | – | | – | | – | |

**Notes.**

F1, Advantages of M-Learning; F2, Limitations in M-Learning; F3, Usability in M-Learning; F4, Freedom in M-Learning.

[+] Present.

[−] Absent.

[*] $P < 0.05$.
**Table 4  Correlations of average daily Internet usage time and age between m-learning attitude scale factors and total score.**

| Factors | F1 | F2 | F3 | F4 | Total score |
|---|---|---|---|---|---|
| Average daily internet usage time | 0.054 (0.639) | 0.243* (0.032) | 0.072 (0.532) | 0.094 (0.413) | 0.138 (0.228) |
| Age | 0.173 (0.131) | 0.192 (0.091) | 0.291* (0.010) | 0.126 (0.270) | 0.229* (0.044) |

Notes.
Correlation coefficient (P).
*$P < 0.05$.

**Table 5  Regression model for total score of the m-learning attitude scale according to purposes of Internet usage.**

| Parameter | $\beta$ | se($\beta$) | $P$-value | 95% CI |
|---|---|---|---|---|
| Online shopping | 1.419 | 3.609 | 0.695 | −5.791 to 8.629 |
| Connecting to social networks | −1.760 | 4.418 | 0.692 | −10.586 to 7.066 |
| Watching videos | −0.502 | 5.784 | 0.931 | −12.058 to 11.053 |
| Reading scientific articles | 9.496 | 7.158 | 0.189 | −4.804 to 23.795 |
| Accessing course materials | 9.397 | 7.158 | 0.588 | −10.403 to 18.196 |
| Checking e-Mails | 8.557 | 9.628 | 0.377 | −10.678 to 27.792 |
| Reading the news | 3.266 | 7.158 | 0.650 | −11.033 to 17.565 |
| Communicating | −7.605 | 3.221 | 0.021* | −14.025 to -1.185 |
| Listening to music | −0.858 | 5.784 | 0.883 | −12.413 to 10.698 |
| Playing games | −12.583 | 9.628 | 0.196 | −31.819 to 6.652 |

Notes.
$\beta$, coefficient; se($\beta$), standard error of coefficient; CI, confidence interval.
*$P < 0.05$ is significant.

mobile technology usage habits of the students were used in the regression model. A stepwise linear regression model (Table 5) included online shopping, connecting to social networks, watching videos, reading scientific articles, accessing course materials, checking e-mails, reading the news, communicating, listening to music, and playing games to predict the total score of the scale, and had a squared multiple correlation coefficient ($R^2$) of 0.268. This result indicated that 27% of the variation in the total score of the m-learning attitude scale according to purposes of Internet usage was explained by only one variable, 'communicating' ($P = 0.021$).

## Opinions about M-Learning

Students were asked whether or not they have applications for educational purposes installed on their mobile phones, and it was found that 28 out of the 81 students (34.56%) used their devices for educational purposes. Students indicated that their most frequently used applications for m-learning were WhatsApp, YouTube, the Dental-lite, foreign language learning, and Atlas of Anatomy.

The students were asked about what the positive aspects of courses with m-learning were, and 31 of the 81 students (38.27%) commented on different features. The most preferred aspects of m-learning were defined as its practicality, and that it was perceived as entertaining, interactive, easily accessible, and repeatable. One student highlighted m-learning daily usage as '*M-learning allows me to better understand the lessons that I missed*

*or hadn't understood at all. It helps me to prepare for exams, and prevents me from carrying around heavy books'*, and another student described m-learning as '*Practical, visual and instructive with animations'*.

On the other hand, although the comments and total score of the m-learning attitude scale showed that the students had a positive attitude toward m-learning; 30 students (37.04%) reported negative aspects as problems associated with the charging of smartphones, distractibility, and Internet connectability problems. Most of the students mentioned issues regarding distraction caused by mobile devices away from learning, with some students saying that '*M-learning may be insufficient where topics are incomprehensible or further details are required'*, '*Faculty readiness should be provided for the use of mobile devices in the classroom',* and '*M-learning may increase the time spent with technological devices. Lectures could become less attractive in the classroom environment. In this regard, our instructors need to renew and adapt their lessons for today's technology'*. These students' approaches suggest that m-learning readiness should be increased, not only on a technological level, but also within the instructional designs.

When students were asked about their expectations for the use of mobile devices in dentistry courses, the students were largely positive towards m-learning. A total of 26 students (32.10%) reported that lectures presented with m-learning activities that utilised more visual materials like videos, animations and photos, were more interesting than traditional forms of teaching with a whiteboard or through PowerPoint presentations. One of the students summarised different features for m-learning as '*Easy to find a lesson, ability to transfer notes between devices, motivating and with a good looking user interface, able to jot down and highlight content, does not slow down your device, resources are open to multifaceted learning, not just based on reading'*. Additionally, students wanted to be able to study at any time with lecture videos or slides that were uploaded to the Internet, and all kinds of documents related to the topics covered on the course that are accessible using mobile devices at any time. Students are also willing to take mobile quizzes in the classroom, and want to be able to take notes on the course materials using their mobile devices.

When asked what kind of content the students wished to see on their mobile devices in order to improve their course success, 28 of the students (34.57%) responded stating courses such as anatomy, dental anatomy and physiology should be supported visually with videos, photos, and applications. The students mentioned a number of application ideas to support their studies interactively such as quizzes, examples of previous exam questions and videos for exam solutions, mobile games on dentistry, and practical applications in dentistry.

## DISCUSSION

Based on their responses, it can be seen that the students are willing and eager to use mobile technologies. As the findings suggest, students use the Internet for an average of 4.49 h per day. The top three Internet communication tools most frequently used by the dentistry students on their devices were online shopping (22.10%), connecting to social networks

(22.10%), and communicating (16.90%). Therefore, the students tend to use mobile technologies for personal use first, followed by educational purposes. The study's findings are similar to those of *Gosper, Malfroy & McKenzie (2013)*, and also *Kukulska-Hulme & Pettit (2009)* in which students' usage of mobile technologies were described. In the study of *Gosper, Malfroy & McKenzie (2013)*, they found that most of the students (over three-quarters) regularly use mobile phones for personal use including text messaging, email, social networking, etc. Even though it is not used as much personal use, the technologies for course related learning activities such as library databases and journals also preferred. *Kukulska-Hulme & Pettit (2009)* also reported that although 96% of the students use mobile phones for social interaction; very few of the students used them for educational purposes (30% for teaching; %17 for learning). A previous study also revealed that the majority of students connect to the Internet on their mobile devices in order to check e-mails (84%) and to use social networks (79%) (*Khatoon, Hill & Walmsley, 2014*). Since the ratio of students with a laptop computer (85.20%) was lower than for a smartphone (97.50%), and almost all of the students had mobile Internet packages (98.80%), the students mostly preferred smartphones for connecting to the Internet. Smartphones and laptops were shown to be popular devices in the current study and in the study of *Khatoon, Hill & Walmsley (2014)*, students reporting usage of a tablet computer (23.50%) showed it to be the less popular electronic device. The results of a study by *Khatoon, Hill & Walmsley (2014)* suggested that smartphone usage may be greater than that of computers due to the latter being more difficult and impractical to carry.

Students reported that they use certain applications for m-learning. The most preferred mobile applications being WhatsApp, YouTube, the Dental-lite, foreign language learning, and Atlas of Anatomy. With regard to mobile applications, similar results were found for dentistry students from different countries (*Chase et al., 2018*; *Deshpande, Kalaskar & Chahande, 2016*; *Khatoon, Hill & Walmsley, 2014*; *Rung, Warnke & Mattheos, 2014*). Dictionary for dental education, applications with quizzes on various lectures such as anatomy and chemistry, and visual tools in dental education were popular purposes for using m-learning applications. It has been recommended that applications developed for a variety of patient cases could improve the clinical decision-making skills of dental students (*Deshpande, Kalaskar & Chahande, 2016*). *Sandholzer et al. (2016)* also revealed that using m-learning apps which are designed to support medical students' knowledge contributes to active student learning. The students preferred to watch videos on YouTube for educational purposes from their smartphones, a finding that was supported by a recent study where students found YouTube videos more helpful than traditional teaching methods (*Khatoon, Hill & Walmsley, 2014*). Similarly, the study of *Botelho, Gao & Jagannathan (2019)* described how video learning supported the knowledge transfer of students with novel learning opportunities.

A previous study showed that students preferred to use smartphones for learning purposes such as taking photographic images of their work, organising timetables for courses, surfing the Internet for other learning materials, and for checking course announcements (*Rung, Warnke & Mattheos, 2014*). In the study of *Gilavand & Shooriabi (2016)*, it was found that smartphones could be used for educational purposes; however,

in various dental faculty curricula, m-learning has not yet been utilised. It is suggested that each educational system could use mobile technologies with appropriate suitable methods, activities and educational materials (*Gilavand & Shooriabi, 2016*; *Taylor et al., 2010*). Specifically, m-learning apps need to be specifically designed for smartphones, as other mobile device types such as tablets may have several disadvantages based on technological aspects and their usability (*Chase et al., 2018*).

The attitudes and views of the dental students are reportedly influenced by many factors. In the study of *Brumini et al. (2014)*, which aimed to measure attitudes towards e-learning amongst dental students, the researchers revealed gender not to be a significant predictor. However, in another study which investigated current knowledge, skills, and opinions of undergraduate dental students with respect to information communication technologies, the skills of male students were found to be better than those of female students, and that they were more familiar in the use of computers, and also longer users of computers (*Rajab & Baqain, 2005*). Despite the study of *Brumini et al. (2014)* and a similar study of *Rajab & Baqain (2005)*, the m-learning attitude scale's total score for males in the current study were found to be statistically significant and higher than that of the females. Moreover, students with portable power supply units were found to reveal statistically higher m-learning attitude scale total scores than those who did not carry such units. The results revealed that students' attitudes using their mobile devices for longer periods could be positively affected by this situation. This finding was also supported by *Patil et al. (2016)* in whose study medical students showed the same interest as dental students.

Another statistically significant result was found on student opinions with regard to the use of mobile devices in dentistry courses. The majority of students (86.9%) thought that the use of mobile devices in dentistry courses was useful and that their attitudes towards m-learning were high. Similar to this result, the study of *Deshpande, Kalaskar & Chahande (2016)* indicated that mobile applications were helpful in learning different aspects of clinical dentistry, and mobile devices could have a significant contribution to modern healthcare education. This result may be due to teaching in dentistry using more visual materials, and therefore students' attitudes towards m-learning in the current study were positive and showed them to be willing users of m-learning. Analysis of the open-ended questions in the current study also confirmed students' perceptions in using multimedia materials such as videos and animations.

There was a positive statistically significant correlation found between the average daily Internet usage and the limitations of m-learning in the current study. Limitations of mobile devices such as charging problems, small screen sizes, and the cost of mobile data significantly affected m-learning (*Dashtestani, 2016*; *Humanante-Ramos, García-Peñalvo & Conde-González, 2017*; *Winters, 2006*). In addition, a significant positive correlation was found to exist between dentistry students' ages and m-learning usability, and the total score of the scale. The increased usage of mobile devices in dentistry over recent years may explain the positive attitude. Similar to the current study's results, a previous study (*Brumini et al., 2014*) showed that higher age demonstrated a significant influence on positive attitudes towards e-learning.

A model constructed to predict the change in m-learning attitude scale total scores according to the purpose of Internet usage showed that only communicating was found to be statistically significant. This finding is consistent with the previous studies of *Khatoon, Hill & Walmsley (2014)* and *Deshpande, Kalaskar & Chahande (2016)* who reported that although social networking and checking e-mails were the most preferred types of communication, instant messaging on smartphones was just as popular. In addition, students used mobile devices as a communication tool for sharing course notes such as presentations, weblinks and photographic images of patients (*Khatoon, Hill & Walmsley, 2014*).

As a limitation of this study, we only focused on the usage habits, attitudes, perceptions, and views of undergraduate dental students towards m-learning, regardless of its efficiency. The faculty has Online Learning Resource Centre which students are able to access course contents over a website of the centre. Therefore, students have experiences to use online resources for educational purposes. We wanted to investigate how they use mobile devices for mobile learning. The study also provides a potential mobile learning project in the future. Our study is limited to one faculty and students from one academic year. Other dentistry faculties in Turkey planning to use m-learning in their education programs may also be included in the new planned studies.

# CONCLUSIONS

To the researchers' knowledge, this is the first study that examines usage habits, attitudes, perceptions, and views of undergraduate dental students towards m-learning in Turkey. According to the data analysis, dental students have generally positive attitudes towards m-learning. They use mobile devices for various purposes and with different habits. The relation between m-learning and mobile device usage was not found to differ significantly. The students raise awareness towards the promises of m-learning in order to utilise their individual technology usage and learning behaviours. Designing learning materials and applications for mobile devices may increase students' overall performances on dental courses. The current study may be seen to contribute to future novel studies relating to m-learning which as a developing need in the field of dentistry education.

## Funding
The authors received no funding for this work.

## Competing Interests
Aslı Suner is an Academic Editor for PeerJ.

## Author Contributions
- Aslı Suner and Yusuf Yilmaz conceived and designed the experiments, performed the experiments, analyzed the data, contributed reagents/materials/analysis tools, prepared figures and/or tables, authored or reviewed drafts of the paper, approved the final draft.

- Beyser Pişkin contributed reagents/materials/analysis tools, authored or reviewed drafts of the paper, approved the final draft.

## Human Ethics

The following information was supplied relating to ethical approvals (i.e., approving body and any reference numbers):

Ege University Scientific Research and Publication Ethics Health Sciences Board granted Ethical approval to carry out the study within its facilities (Ethical Application Ref: 145-2018).

## Data Availability

The raw measurements are provided in Data S1.

## Supplemental Information

Supplemental information for this article can be found online at http://dx.doi.org/10.7717/peerj.7391#supplemental-information.

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
