# Peer review of "Mobile learning in dentistry: usage habits, attitudes and perceptions of undergraduate students"

_PeerJ, doi:10.7717/peerj.7391_

## Round 0.1 · original submission · Minor Revisions

Your manuscript addresses a relevant modern educational trend among dental students. Please address the reviewer comments on a point by point basis and make the suggested changes before resubmitting.

Reviewer 1 ·

Basic reporting

The study has spelling that is not correct. Lines: 53...programmes, line 267...utilised. And others.
References cover the domains of content. they are well based.
Hypotheses are correct and valid, tables support them.

Experimental design

Well done

Validity of the findings

No comment

Additional comments

This is important information for educators as schools plan for future efficiencies in learning and success in life long learning models and vehicles.

·

Basic reporting

No comment

Experimental design

No comment

Validity of the findings

Please describe the independent groups compared and the study population (see description below in comments).

Additional comments

This article presents an interesting study to assess mobile learning in undergraduate dental students. Please address the following comments to improve this manuscript:
• IRB approval-Add reference number
• In the abstract, under Materials and Methods, it is listed that total scores of two independent groups were compared with Mann Whitney test and of three independent groups with Kruskall-Wallis test. These respective independent 2 and 3 groups are not described in the manuscript. Rather, the Materials and Methods section describes that no sampling method was used. Please describe how these groups were selected.
• Under Introduction, paragraph 2 “In addition, whilst it is possible for students…..”, change ‘effectively’ to ‘effective’ as it is intended to be used as an adjective and not an adverb
• In Materials and Methods, it is mentioned that “the study was designed as a cross-sectional survey (2014)” while the study is conducted in 2017-18. Please clarify if the survey was designed in 2014, if so, was it published then and add a reference to that publication.
• The study population, undergraduate dental students, should be described in detail, specifically if these were 1st, 2nd, 3rd or 4th year students as from the quoted responses, it seems the survey was taken by the junior dental students only.
• It should also be discussed if the dental curriculum in the study institution has an online educational system or curriculum management system. The conclusion that the students use internet primarily for shopping, social networks and communication, and secondary for educational purposes might not be accurate if they are not introduced to an internet-based educational system in the first place.

Reviewer 3 ·

Basic reporting

Overall, this is an interesting study and it seems like it offers something new to what is already known about m-learning in dental education. I believe it would benefit greatly from attention to several points that I have outlined in my comments.

1. There are multiple areas where the language needs to be corrected or the read-ability of a sentence could be improved. I have noted specific instances of these and offered suggestions below.

2. The review of the literature is generally well done, however the introduction would benefit from the addition of a paragraph outlining what is currently known to be associated with attitudes towards m-learning.

3. The paper has been structured well, and the flow of materials is logical.

4. The results reported fit well with the stated study aims.

5. I have the following comments for your introduction:
-line 38 – write out “IT” in full as this is the first time you use this abbreviation
-line 41 – I find the use of the word “solutions” ambiguous. Consider rephrasing to read as “…educational technologies have taken advantage of this opportunity.” (if I am understanding your intent with this sentence correctly).
-line 47 – “It is emphasized …” I would suggest making the source of the emphasis explicit in this sentence and re-phrasing it to read something like this: “It is emphasized in the m-learning literature that …” or “Results of m-learning research emphasize that …”
-line 50 – I would suggest qualifying your description of courses as “intensive” – do you mean they are time-intensive, knowledge-intensive, or in a different way intensive? I think that specifying this will help strengthen your argument for why m-learning is so important.
-line 53-54 – Consider re-wording this sentence to improve clarity and grammar. For example, “In addition, whilst it is possible for students to withstand intensive course content delivery over a short period of time, prolonged instruction at this intensity may make meaningful learning more difficult (reference).”

-paragraph starting at line 50: Consider adding an opening sentence and several linking words to summarize the content of this paragraph. For example, “There are multiple advantages that m-learning brings to education. First … Second …. Third ….” This would make the content in this paragraph easier for the reader to follow.

Experimental design

1. This paper appears to fit with the stated aims and scope of this journal.

2. I think your research aim could be better structured. Consider stating the study’s general purpose and then following that up with specific study objectives. For example, “Therefore, the aim of the current study was to determine the mobile technology usage habits of 81 undergraduate dental students and their attitudes and opinions towards m-learning” [study purpose]. Specifically, the study objectives were as follows: (1) to describe usage of mobile technology in dental students, (2) to explore students attitudes towards m-learning, and (3) to explore the role of the students’ demographic information and the purpose and habits of using mobile technologies as examined by previous research 90 studies (Deshpande, Kalaskar, & Chahande, 2016; Khatoon et al., 2014; Rung, Warnke, & 91 Mattheos, 2014). Or, state the study objectives using the breakdown detailed in lines 153-159.

3. You have included mention of ethical review as requested by the journal.

I think your methods require greater detail and some of your decisions require support as outline below:
-lines 94-95 – You have repeated the study’s purpose here. Consider revising to read as follows “To address the study’s aim, a cross-sectional survey design was utilized.”
-ethical approval is included

-line 99 – “The study consists of 81 volunteer participating undergraduate students … would read better as “The sample consisted of 81 undergraduate students who volunteered to participate …”

M-learning Attitude Scale
-line 104: Omit “with” and change to past tense (consists to consisted)
-line 108 – to help the reader, specify what a higher score indicates (ie. a better or worse attitude towards m-learning)
-please specify the scales here

Student Information Form
–More detail on this item or an example of an item would benefit the reader. For example, can an example of an item that captured current usage be included? Was a student asked about usage that day or in that term or in the past 2 weeks?

Opinions about mobile learning
–include the 5 questions here

Statistical analysis
-line 143 – This is the first mention of two independent groups. This needs to be described earlier in the methods, because the reader does not know what criteria was used to divide the 81 students into these two groups.
-it is unclear why you decided to only include reasons for internet usage in your regression analysis – why not include sociodemographic information?

Validity of the findings

1. You have provided your dataset as requested by the journal.

2. I have the following comments for your results section:
-Line 186 – “students” should be “students’ ”
-line 229 – Interpretation of the r square would help readers understand your results. For example, 27% of the variation in … was explained by ….(r2=0.268).”
-line 238 – Consider making this sentence flow better by re-wording as follows “The students were asked about positive aspects of courses with m-learning,…”

3. I have the following comments for your tables:
Table 1
-specify that “M” stands for the median as your tables should be able to stand alone from your discussion of statistical methods
-including the possible range of scores would also help provide context for the data you present here

Table 2
-because usage habits are not described in the methods, I am unclear as to what the total score refers

Table 3
-a note about the different factors here would be helpful as otherwise it is unclear as to what “F1” refers without having to go back and check your methods

4. I have the following comments for your discussion:
-lines277-278 – “…are very much into …” Consider revising to more professional language such as “…are willing and eager to use …”
-lines 282-283 – Please specify the findings of this reference – did the other authors also identify the same trend of personal use first and educational use second?
-line 292 – Change “reported” to “reporting”
-line 293-295 has multiple grammatical issues. Consider revising to “The results of a study by Khatoon et al. (2014) suggest that smartphone usage may be greater than that of computers due to the latter being more difficult and impractical to carry.”
-line 305 “Contributes” should be “contribute”
-line 317 – Select one of “suitable” or “appropriate” but use of both words here is redundant

-a discussion of the limitations of the study is missing and needs to be added

-consider including an implications paragraph

Reviewer 4 ·

Basic reporting

Please see comments below.

Experimental design

Please see comments below.

Validity of the findings

Please see comments below.

Additional comments

General comments
=============
This manuscript aims to evaluate attitudes and perceptions towards mobile learning
(m-learning), as well as to identify variables related to those attitudes amongst undergraduate dental
students.. I would like to congratulate the authors for their efforts in planning the research and writing this manuscript.

Specific comments
=============

Major comments
* * *
1. I understand that the study population consisted of volunteers (n = 81) and the data collection was carried out through a virtual survey. Have you considered the fact that these students (due to the fact of voluntariness), have already had some ease in handling social media and other internet-related technologies? This could explain why the authors found, in general, positive attitudes. I have not seen this in the Discussion section (especially in the limitations).
2. Results section: Some data presented in the tables are unnecessarily repeated in the text. For example, lines 172-173, 175-177. Please correct.

Minor comments
* * *
1. Line 38: Please write the full term of "IT", and then use the abbreviation in parentheses.
2. Line 88 ("These variables are...") to line 92: This text is probably unnecessary as it is already presented in the Methods section
3. Lines 279-280 have the same information that lines 284-286.
4. Line 291: Why the sencence starts with "although"?

---

## Round 0.2 · accepted · Accept

Thanks for addressing the reviewer responses.

Reviewer 4 ·

Basic reporting

Please see comments below.

Experimental design

Please see comments below.

Validity of the findings

Please see comments below.

Additional comments

The authors have addressed all the suggestions. I believe the manuscript is now suitable for its publication.